# *Porphyromonas gingivalis* Lipopolysaccharides Promote Proliferation and Migration of Human Vascular Smooth Muscle Cells through the MAPK/TLR4 Pathway

**DOI:** 10.3390/ijms24010125

**Published:** 2022-12-21

**Authors:** Megumi Miyabe, Nobuhisa Nakamura, Tomokazu Saiki, Satoru Miyabe, Mizuho Ito, Sachiko Sasajima, Tomomi Minato, Tatsuaki Matsubara, Keiko Naruse

**Affiliations:** 1Department of Internal Medicine, School of Dentistry, Aichi Gakuin University, 2-11, Suemori-dori, Chikusa-ku, Nagoya 464-8651, Japan; 2Department of Pharmacy, Aichi Gakuin University Dental Hospital, 2-11, Suemori-dori, Chikusa-ku, Nagoya 464-8651, Japan; 3Department of Maxillofacial Surgery, School of Dentistry, Aichi Gakuin University, 2-11, Suemori-dori, Chikusa-ku, Nagoya 464-8651, Japan; 4Department of Clinical Laboratory, Aichi Gakuin University Dental Hospital, 2-11, Suemori-dori, Chikusa-ku, Nagoya 464-8651, Japan; 5The Graduate Center of Human Sciences, Aichi Mizuho College, Syunko-cho, Mizuho-ku, Nagoya 467-0867, Japan

**Keywords:** atherosclerosis, vascular smooth muscle cell, inflammation

## Abstract

Atherosclerosis is a major cause of mortality worldwide. The initial change in atherosclerosis is intimal thickening due to muscle cell proliferation and migration. A correlation has been observed between periodontal disease and atherosclerosis. Here, we investigated the proliferation and migration of human aortic smooth muscle cells (HASMCs) using *Porphyromonas gingivalis*-derived LPS (*Pg*-LPS). To elucidate intracellular signaling, toll-like receptor 4 (TLR4) and myeloid differentiation factor 88 (MyD88) of HASMCs were knocked down, and the role of these molecules in *Pg*-LPS-stimulated proliferation and migration was examined. The role of mitogen-activated protein kinase (MAPK) in HASMC proliferation and migration was further elucidated by MAPK inhibition. *Pg*-LPS stimulation increased the proliferation and migration of HASMCs and activated the TLR4/MyD88 pathway. TLR4 knockdown inhibited *Pg*-LPS stimulated HASMCs proliferation and migration. *Pg*-LPS stimulation led to the phosphorylation of P38 MAPK, JNK, and ERK, and MyD88 knockdown inhibited the phosphorylation of P38 MAPK and JNK but not ERK. P38 MAPK and SAPK/JNK inhibition did not suppress the proliferation of HASMCs upon *Pg*-LPS stimulation, but ERK inhibition significantly inhibited proliferation. SAPK/JNK and ERK inhibition suppressed *Pg*-LPS-stimulated migration of HASMCs. In conclusion, our findings suggest that *Pg*-LPS may promote atherosclerosis via the activation of MAPK through TLR4.

## 1. Introduction

Atherosclerotic vascular diseases such as coronary vascular disease are life-threatening illnesses that affect many people worldwide [1]. Proliferation and migration of vascular smooth muscle cells (SMCs) are deeply involved in neointimal formation and vessel stenosis during vascular remodeling [2,3,4]. Elucidation of the factors that regulate the response of SMC may help prevent and/or treat angiopathy. Additionally, research reports showing that inflammation is involved in arteriosclerosis provide a new direction for elucidating the mechanism of cardiovascular disease caused by atherosclerosis and for the development of therapeutic methods [5,6]. Furthermore, it has been proposed that arteriosclerosis begins when T cells and macrophages adhere to and collect on the injured vascular endothelium and foam cells and is regarded as a chronic inflammatory disease [7,8,9]. Severe periodontitis is more likely to be associated with cardiovascular disease [7,8,9] and animal models of atherosclerosis have demonstrated increased atherosclerosis in those infected with the periodontopathic bacterium *Porphyromonas gingivalis* (*Pg*) [10]. Additionally, many epidemiological studies have reported that periodontal disease, a chronic inflammatory disease, is associated with arteriosclerotic cardiovascular disorders [7,8,9,11,12]. However, the detailed mechanism remains unclear.

The etiology of periodontitis is related to the destruction of periodontal tissue caused by the persistent infection of the periodontal pockets between teeth and gums by periodontopathic bacteria [13,14]. *Pg* is the most frequently isolated bacterium from the inflammatory site of periodontitis in adults. It has virulence factors such as lipopolysaccharide (LPS) and fimbriae and toxic factors such as gingipine and collagenase [15]. 

The mechanism by which periodontal disease is involved in arteriosclerosis is still unclear. Previous reports have shown that cell proliferation is greatly increased when cells are stimulated with *Escherichia coli*-derived LPS (*E. coli*-LPS) [16]. Although *E. coli* is a resident bacterium, it is rarely detected in the oral cavity of healthy individuals [17] and is not a major cause of periodontal diseases. *Pg* is a major cause of periodontal disease [18,19,20]. It has also been reported that arteriosclerosis was accelerated after oral administration of *Pg* bacteria to arteriosclerosis model mice [21,22]. Since infection with *Pg* bacteria results in exposure to LPS-derived from *Porphyromonas gingivalis* (*Pg*-LPS), we hypothesized that *Pg*-LPS affects vascular cells to promote atherosclerosis. In this study, we investigated the effects of *Pg*-LPS on the proliferation and migration of human aortic smooth muscle cells (HASMCs) and the underlying molecular mechanisms.

## 2. Results

### 2.1. Pg-LPS Stimulates HASMC Proliferation and Migration

To investigate the effect of *Pg*-LPS on cell proliferation, HASMCs were stimulated with *Pg*-LPS at different concentrations (1, 10, and 100 ng/mL) for 16 h. MTT assays showed that *Pg*-LPS stimulated the number of HASMCs in a dose-dependent manner (Figure 1A). The DNA synthesis level in HASMCs was measured using a BrdU incorporation assay. *Pg*-LPS increased DNA synthesis in a dose-dependent manner (Figure 1B). The number of HASMCs and DNA synthesis significantly increased after *Pg*-LPS (100 ng/mL) stimulation compared to that in the control group.

Scratch wound healing and transwell assays were performed to examine the effects of *Pg*-LPS on HASMC migration. In the scratch wound healing assay, *Pg*-LPS (100 ng/mL) significantly stimulated HASMC migration (Figure 1C). Similarly, the transwell assay revealed that stimulation with *Pg*-LPS increased the number of migrating HASMCs (Figure 1D). These results indicate that *Pg*-LPS increases the proliferation and migration of HASMCs.

### 2.2. Pg-LPS Activates the TLR4/MyD88 Signaling Pathway of HASMCs

To clarify the intracellular signals by which *Pg*-LPS stimulated HASMC proliferation and migration, we investigated the expression of toll-like receptor 4 (TLR4), a receptor for LPS. Subsequently, we examined the expression of myeloid differentiation factor 88 (MyD88), a TLR4 adapter molecule. The TLR4 and MyD88 protein levels were assessed using Western blot analysis. *Pg*-LPS stimulation enhanced TLR4 and MyD88 expression (Figure 2A). Quantitative analysis revealed that TLR4 and MyD88 protein expression increased in a time-dependent manner, with maximal levels occurring 3 h after *Pg*-LPS stimulation (Figure 2B). These results indicate that the treatment of HASMCs with *Pg*-LPS activates the TLR4/MyD88 signaling pathway.

### 2.3. TLR4 siRNA Suppresses Proliferation and Migration of HASMCs 

It has been reported that the effects of *E. coli*- LPS on SMCs are mediated by TLR4 [23,24]. Our results suggested that the proliferation and migration of HASMCs stimulated by *Pg*-LPS are also mediated by TLR4. To clarify the role of TLR4 in HASMC proliferation and migration, TLR4 in HASMCs was knocked down using siRNA against TLR4, which was transfected using HiPerFect Transfection Reagent. The silencing effect was confirmed using Western blot analysis. TLR4 protein levels were suppressed compared to the control (Appendix A). After TLR4 was knocked down in HASMCs (siRNA at 5 nM), the cells were stimulated with *Pg*-LPS for 16 h. TLR4 suppression abolished the *Pg*-LPS-stimulated increase in cell number and DNA synthesis in HASMCs (Figure 3A,B). The scratch wound assay showed that TLR4 knockdown significantly suppressed wound closure following *Pg*-LPS stimulation (Figure 3C). Similarly, the transwell assay showed that the number of cells stimulated by *Pg*-LPS was significantly suppressed by TLR4 knockdown (Figure 3D). These results indicated that *Pg*-LPS stimulates HASMC proliferation and migration via TRL4.

### 2.4. MyD88 siRNA Suppresses the Migration of HASMCs

MyD88 has been reported to induce cellular responses in various cells via TLR4/MyD88/MAPK or NF-κB. Therefore, in this study, we knocked down MyD88 and investigated the proliferation and migration of HASMCs induced by *Pg*-LPS stimulation. MyD88 in HASMCs was knocked down using siRNA against MyD88, which was transfected using the HiPerFect Transfection Reagent. The silencing effect was confirmed using Western blot analysis. MyD88 protein levels were suppressed compared to the control (Appendix A). After MyD88 was knocked down in HASMCs (siRNA at 5 nM), the cells were stimulated with *Pg*-LPS for 16 h. Knockdown of MyD88 tended to suppress the *Pg*-LPS-stimulated increase in cell number and DNA synthesis in HASMCs, but these effects were not significant (Figure 4A,B). The scratch wound assay showed that MyD88 knockdown significantly suppressed wound closure following *Pg*-LPS stimulation (Figure 4C). Similarly, the transwell assay showed that the number of cells stimulated by *Pg*-LPS was significantly suppressed by MyD88 knockdown (Figure 4D). These results indicated that *Pg*-LPS promoted HASMC migration, but not proliferation, through MyD88.

### 2.5. TLR4/MyD88 Mediates the Pg-LPS-Induced Activation of p38 MAPK and SAPK/JNK, but ERK Activation Is Independent of MyD88

To elucidate the signaling pathways involved in *Pg*-LPS-induced cell proliferation and migration of HASMCs, we investigated the phosphorylation of p38 mitogen-activated protein kinase (p38 MAPK) and stress-activated protein kinase/Jun-amino-terminal kinase (SAPK/JNK), which are TLR4/MyD88 downstream target molecules. As p44/42 MAPK (ERK) activation is essential for HASMC proliferation and migration [25], the phosphorylation levels of p38 MAPK at Thr180/Tyr182, SAPK/JNK at Thr183/Tyr185, and ERK at Thr202/Tyr204 were evaluated by Western blot analysis (Figure 5A). Quantitative evaluation of the p38 MAPK, SAPK/JNK, and ERK phosphorylation levels showed that phosphorylation levels reached their peak only 5 min after *Pg*-LPS stimulation and then gradually declined (Figure 5B). Our results demonstrated that P38MAPK, SAPK/JNK, and ERK in HASMC were activated by *Pg*-LPS stimulation.

To clarify the relationship between TLR4 and p38 MAPK, SAPK/JNK, and ERK in HASMCs, TLR4 was knocked down using siRNA (siRNA at 5 nM). Quantitative evaluation of the phosphorylation levels revealed that TLR4 knockdown significantly attenuated the phosphorylation levels of p38 MAPK, SAPK/JNK, and ERK stimulated by *Pg*-LPS (Figure 6A,B). Subsequently, to investigate the relationship between MyD88 and MAPK, we knocked down MyD88 (siRNA at 5 nM) in HASMCs and evaluated the phosphorylation of p38 MAPK, SAPK/JNK, and ERK by Western blot analysis (Figure 6C). Quantitative evaluation of phosphorylation levels revealed that MyD88 knockdown significantly attenuated the phosphorylation levels of p38 MAPK and SAPK/JNK stimulated by *Pg*-LPS but not ERK phosphorylation (Figure 6D). Collectively, these results indicate that stimulation of HASMCs by *Pg*-LPS activates P38MAPK and SAPK/JNK via the TLR4/MyD88 signaling pathway; however, *Pg*-LPS-induced ERK activation is mediated by TLR4 and is independent of MyD88.

### 2.6. MAPK Signaling Attenuates Pg-LPS-Induced Migration of HASMCs but Not Proliferation

Our results indicated that the knockdown of TLR4 in HASMCs suppressed the activation of MAPK signaling, including ERK, and suppressed *Pg*-LPS-induced proliferation and migration. Additionally, the knockdown of MyD88 in HASMCs suppressed the activation of p38 MAPK and SAPK/JNK. To elucidate the signaling pathway, we investigated the effects of inhibitors of each pathway on the *Pg*-LPS-stimulated HASMC proliferation and migration. The cell number and DNA synthesis of HASMC stimulated by *Pg*-LPS were significantly suppressed in the PD98059 group (an ERK inhibitor), but not in the SB203580 group (a p38 MAPK inhibitor) or SP600125 group (an SAPK/JNK inhibitor) (Figure 7A,B). The scratch wound assay showed significant wound closure suppression following *Pg*-LPS stimulation in the PD98059, SB203580, and SP600125 groups (Figure 7C). The transwell assay showed that the number of migrating cells stimulated by *Pg*-LPS was suppressed in the PD98059 and SP600125 groups but not in the SB203580 group (Figure 7D). These results indicate that ERK inhibition suppresses *Pg*-LPS-induced proliferation and migration of HASMCs, whereas p38 MAPK and SAPK/JNK inhibition suppress migration but not proliferation.

## 3. Discussion

In this study, we evaluated the proliferation and migration of HASMCs stimulated by *Pg*-LPS, which made it possible to evaluate the mechanism of atherosclerosis development due to the presence of *Pg*-LPS. Our results indicated that *Pg*-LPS stimulation increased the proliferation and migration of HASMCs. 

Atherosclerotic plaques are composed of various cell populations, including vascular endothelial cells, vascular SMCs, macrophages, and lymphocytes. Ross et al. proposed that vascular endothelial cell damage leads to inflammatory changes that induce macrophage and lymphocyte invasion of the vessel wall, secretion of growth factors, cytokines, and chemokines, and transformation of vascular SMCs [6]. Vascular intimal thickening is believed to be caused by the proliferation and migration of dedifferentiated muscle cells [2,26,27]. ERK and p38 MAPK are involved in SMC dedifferentiation [28]. Pathological analysis has also indicated that intimal thickening due to the proliferation and migration of vascular SMCs is the main factor in arteriosclerosis development [29]. Here, we demonstrated that *Pg*-LPS increased HASMC proliferation and migration, suggesting that the presence of *Pg*-LPS promotes arteriosclerosis. 

When evaluating cell proliferation in studies on atherosclerosis, some studies reported 20 or 24 h from cell stimulation to evaluation [30,31], while others reported significance in cell DNA synthesis at 6 h or more of *E. coli*-LPS stimulation [16]. We evaluated several stimulation times and found that HASMCs proliferate and migrate at 16 h after stimulation, similar to the previous reports. In our experiments, *Pg*-LPS stimulation also significantly increased cell proliferation and migration, but to a lesser extent than *E. coli*-LPS stimulation (data not shown). It is considered that *Pg*-LPS stimulation has a weaker effect on cells than *E. coli*-LPS stimulation because of the lower expression of adhesion molecules in HUVECs with *Pg*-LPS stimulation than with *E. coli*-LPS stimulation [32]. 

*Pg*-LPS-induced TLR4 and MyD88 protein expression peaked 3 h after stimulation initiation. A previous report on *E. coli*-LPS stimulation showed that TLR4 mRNA expression peaked 2 h after stimulation [33]. Other studies have compared TLR4 and MyD88 protein expression 2 h after *E. coli*-LPS stimulation [23]. Therefore, protein expression of TLR4 and MyD88 by *Pg*-LPS stimulation was evaluated from 0–240 min. The time difference in the TLR4 and MyD88 peak expression may be due to the weaker effect of *Pg*-LPS compared with the effect of *E. coli*-LPS, as mentioned earlier.

We demonstrated that *Pg*-LPS-induced HASMC proliferation was mediated by the TLR4/MyD88 pathway. On the other hand, *Pg*-LPS-induced HASMC migration was mediated by TLR4, but not MyD88. We also investigated the activation of P38MAPK and SAPK/JNK, which are TLR4 and ERK downstream molecules and are deeply involved in cell proliferation [34]. It has been reported that *Pg*-LPS stimulation activates MAPK in human gingival fibroblasts and periodontal ligament stem cells [35,36]. In this study, we found that *Pg*-LPS activated P38MAPK, SAPK/JNK, and ERK in HASMCs. Furthermore, it has been reported that LPS-stimulated ERK activation is MyD88 dependent or independent [35,37]. Here, we evaluated the activation of MAPKs (p38 MAPK, SAPK/JNK, and ERK) by TLR4 and MyD88 knockdown in HASMCs. Our results indicated that P38MAPK and SAPK/JNK were activated via TLR4/MyD88 in HASMCs after *Pg*-LPS stimulation, whereas ERK was activated via TLR4 after *Pg*-LPS stimulation in a MyD88-independent manner. Furthermore, our results showed that P38MAPK and SAPK/JNK inhibition did not suppress HASMC proliferation upon *Pg*-LPS stimulation, whereas ERK inhibition significantly inhibited proliferation. SAPK/JNK and ERK inhibition suppressed the migration of *Pg*-LPS-stimulated HASMCs. P38MAPK inhibition tended to inhibit migration, but the inhibitory effect was weaker than that of SAPK/JNK and ERK. Proliferation and migration of dedifferentiated SMCs are essential for the initial changes in atherosclerosis [6,29], and p38 MAPK and JNK are involved in the dedifferentiation of SMCs [34,38]. Furthermore, activation of P38MAPK, JNK, and ERK is involved in SMC migration [39,40] and activation of JNK and ERK is involved in its proliferation [40]. Our results indicate that in HASMCs stimulated with *Pg*-LPS, phosphorylation of p38 MAPK and SAPK/JNK via TLR4/MyD88 promotes migration, while MyD88-independent ERK phosphorylation via TLR4 promotes both proliferation and migration (Figure 1).

## 4. Materials and Methods

### 4.1. Reagents

TLR4 antibody was purchased from Santa Cruz Biotechnology (Santa Cruz, CA, USA). Antibodies for P38 MAPK, SAPK/JNK, ERK, and MyD88, and anti-rabbit and anti-mouse secondary antibodies, were purchased from Cell Signaling Technology (Danvers, MA, USA). *Pg*-LPS was purchased from in vivo Gen (San Diego, CA, USA).

### 4.2. Cell Culture and Treatment 

HASMCs were purchased from KURABO Industries (Osaka, Japan). These cells are normal human aortic smooth muscle cells, primary cells from a Caucasian 46-year-old male. HASMCs were cultured in Humedia SG-2 (KURABO Industries, Osaka, Japan), as recommended by the manufacturer, and cells from passages 4–7 were used for the experiments. HASMCs were incubated in DMEM containing 0.2% fetal bovine serum (FBS) for 24 h before the experiments. In some experiments, cells were treated with inhibitors of MAPK, p-38, and JNK for the indicated times.

### 4.3. Proliferation Assays 

The cell number was assessed using the Cell Proliferation Kit I (MTT) (Roche Diagnostics, Basel, Switzerland). DNA synthesis was measured using a BrdU proliferation assay kit (Roche Diagnostics, Basel, Switzerland), according to the manufacturer’s instructions [38]. HASMCs were stimulated with *Pg*-LPS for the indicated periods.

### 4.4. Cell Migration Assays 

Directional cell migration of HASMCs was stimulated using an in vitro scratch wound healing assay. First, HASMCs (5 × 10^5^) were seeded in 6-well plates and cultured in Humedia-SG2. After reaching confluency, the cells were serum-starved for 24 h and directly scratched with a 200 μL pipette tip. The cells were then incubated with or without *Pg*-LPS (100 ng/mL) for 16 h. The wound healing area was assessed by measuring the distance of the wound edge using the ImageJ software. The migratory properties of HASMCs were assessed by transwell assay with polycarbonate membranes coated with fibronectin [41]. HASMCs were added to the upper chamber, and serum-deprived medium supplemented with *Pg*-LPS was added to the lower chamber. The cells were allowed to migrate through the membrane pores for 16 h.

### 4.5. Western Blot Analysis 

Cell samples were homogenized in radioimmunoprecipitation assay (RIPA) buffer (Cell Signaling Technology, Danvers, MA, USA) containing 1 mM phenylmethylsulfonyl fluoride (PMSF) (Sigma-Aldrich Co. LLC, St. Louis, MO, USA). Equal amounts of protein were subjected to SDS-PAGE and transferred to PVDF membranes. Membranes were incubated with the indicated antibodies, followed by incubation with a secondary antibody.

### 4.6. Knockdown of Gene Expression with Interference RNA

Intracellular TLR4 and MyD88 expression were knocked down by transfection with interference RNA (siRNA). The negative control siRNA was AllStars Negative Control siRNA (Quiagen, Venlo, The Netherlands), a non-silencing siRNA. The siRNA was transfected into HASMCs using a HiPerFect Transfection Reagent (Qiagen, Venlo, The Netherlands), according to the manufacturer’s instructions.

### 4.7. Statistical Analysis

Data are presented as the mean ± SE. An analysis of variance (ANOVA) with the Tukey-Kramer honestly significant difference test or Student’s t-test was used to compare the difference in the mean among the groups when the data approximately represented a normal distribution. A *p*-value < 0.05 indicated a statistically significant difference.

## 5. Conclusions

Our study shows that *Pg*-LPS stimulation promotes HASMC migration via TLR4/MyD88-mediated phosphorylation of p38 MAPK and SAPK/JNK, while MyD88-independent ERK phosphorylation via TLR4 promotes HASMC proliferation and migration. These results suggest that *Pg*-LPS may promote arteriosclerosis. Approaches targeting periodontitis will effectively prevent atherosclerosis progression and are a subject for future research.

## Data Availability

The data presented in this study are available on request from the corresponding author.

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
