# Peer review of "Porphyromonas gingivalis Lipopolysaccharides Promote Proliferation and Migration of Human Vascular Smooth Muscle Cells through the MAPK/TLR4 Pathway"

_ijms, 2022, doi:10.3390/ijms24010125_

Round 1

Reviewer 1 Report

The paper by Miyabe et al. entitled “Porphyromonas gingivalis lipopolysaccharides promote proliferation and migration of human vascular smooth muscle cells through the MAPK/TLR4 pathway” aims to investigate how periodontal disease may cause and promote atherosclerosis. Overall, the work is clear, linear, simply described. For each experiment performed the rational is well described. The introduction is short but sufficient. I have only some comments:

Line 26: typos (differen-tiation)

Lines 47-49: confused, please rephrase

If proliferation ad BrdU assay have been performed at 16h after pg-LPS stimulation, why WB of TLR4 and MyD88 have bene done between 0 to 240 minutes? What about the expression of these proteins are after 16h of stimulation? The same question rises up for WB in figure 4.

The WB of p38 in figure 4A is not so representative. Indeed, seeing the low level of expression of t38 at time 0, it is not so clear the increase of p38/t38 ratio.

Reviewer 2 Report

IJMS:

Manuscript ID: ijms-2042146
Title: Porphyromonas gingivalis lipopolysaccharides promote proliferation and migration of human vascular smooth muscle cells through the MAPK/TLR4 pathway

Authors: Megumi Miyabe et al.

Brief summary:

This study investigates the effects of LPS isolated from the periodontopathic bacterium Porphyromonas gingivalis (Pg) on proliferation and migration of human aortic smooth muscle cells (HASMC) using proliferation, wound scratch, and transwell assays. Knockdown of TLR4 and MyD88, combined with pharmacological MAPK inhibition, dissect the downstream signaling mechanisms involved in Pg-induced HASMC migration and proliferation. The authors conclude that periodontal disease may cause and promote atherosclerosis via TLR4/MyD88/MAPK-dependent pathways.

Strengths:

-        Use of human cells

-        Combination of genetic and pharmacological approaches

-        Logically-designed experiments

Main concern:

The findings of this study do neither replicate a model of periodontal disease nor of vascular atherosclerosis, which is the backdrop this study is build on and a major conclusion reached by the authors. Similarly, the presented data do not establish a mechanistic link between periodontitis and atherosclerosis, as proposed by the authors throughout the manuscript. Instead, the reported experimental set-up reflects the acute (16h) effects of Pg-LPS on proliferation and migration of human aortic smooth muscle cells (HASMC), and the study should simply be framed as such. Furthermore, as the authors state, periodontal disease is a chronic disease, whereas the proposed experimental setup (16 hrs of LPS) reflects acute stimulation.

Nevertheless, the experiments underlying the presented data are for the most part actually well-designed, but -as mentioned- do not allow for drawing any conclusions about periodontal disease, atherosclerosis, and the mechanistic relationship between these two disease processes. Therefore, my suggestion is to rewrite the manuscript with keeping this concern in mind, and introduce and describe the findings simply in the context of LPS-induced cell proliferation/migration. In the Discussion section the authors could then speculate how these findings could be relevant for diseases such as periodontitis and atherosclerosis.

Additional concerns/questions:

--It would significantly strengthen the manuscript if the author could replicate at least key findings with the actual live bacterium (Pg), not just LPS.

--Although both TLR4 and MyD88 expression increase after Pg-LPS (Fig. 2), no studies are shown that establish a causal link between these two molecules to support the authors’ conclusion that TLR4 activates MyD88. Similarly, proliferation and wound healing studies using MyD88 siRNA-treaded HASMC are lacking to support the conclusion that MyD88 regulates proliferation and wound healing.

--The data showing TLR4 siRNA knockdown should be included and quantified (instead of reported as “significantly suppressed”), as major conclusions are drawn from this experimental setup.

--source and phenotype characterization of HASMC should be reported, and references listed. In addition, are these primary or transformed cells?

--what transfection controls were used for the siRNA assays, e.g. a scrambled peptide? These are important controls and should also be shown. Commercial source and siRNA concentrations are lacking.

--In Fig. 1A/B, is there a dose-effect between the different LPS concentrations?

--A simple graphic summary of the results would be very helpful to provide a visual overview of the proposed signaling pathways and outcomes.
